# Intronic *SYNE1* Gene Novel Variant Associated with Myocardial Infarction in Young People with a Family History of Premature Atherosclerosis: A Case–Control Study in the Polish Population [note 1]

**DOI:** 10.3390/ijms26052244

**Published:** 2025-03-03

**Authors:** Michał Ambroziak, Jakub Franke, Anna Wójcicka, Monika Kolanowska, Tomasz Jaxa-Chamiec, Andrzej Budaj

**Affiliations:** 1Department of Cardiology, Centre of Postgraduate Medical Education, Grochowski Hospital, 04-073 Warsaw, Poland; tomjch@kkcmkp.pl (T.J.-C.); abudaj@kkcmkp.pl (A.B.); 2Warsaw Genomics, 02-502 Warsaw, Poland; jakub.t.franke@gmail.com (J.F.); amzwojcicka@gmail.com (A.W.); monika.kolanowska@gmail.com (M.K.); 3II Department of Radiology, Medical University of Warsaw, 02-097 Warsaw, Poland; 4Fundacja Wiedziec Wiecej, 01-682 Warsaw, Poland; 5Department of Tumor Biology and Genetics, Medical University of Warsaw, 02-106 Warsaw, Poland

**Keywords:** *SYNE1*, myocardial infarction, myocardial infarction at young age, coronary artery disease, premature coronary artery disease

## Abstract

Premature myocardial infarction (MI) risk factors, including genetic ones, are crucial for an individual risk stratification. The aim of this study was to investigate the role of genetic variants in young patients with MI and a family history of premature atherosclerosis (FHpa). The studied group consisted of 70 patients aged 26–49 (mean 43.1, SD ± 4.3; 17 women, 53 men), with MI and with FHpa. The targeted enrichment library was prepared and analyzed using the Next-Generation Sequencing method. The results of sequencing were compared to data from the reference control population, consisting of 597 people with no history of MI (418 women, 179 men) aged 18–83 (mean 40.5, SD ± 12.4), using Propensity Score Matching. *SYNE1* gene variant NM_182961.4:c.20396+22A>G occurs with a significantly higher incidence in the studied group compared to the control population (OR 4.80 95%CI 1.43–14.45; *p* = 0.005) as a whole and when matched by age and gender (OR 9.31 95%CI 1.64–95.41; *p* = 0.004). There were no statistically significant differences in the incidence of variants related to familial hypercholesterolemia (*LDLR* NM_001195800.2:c.667G>A, *PCSK9* NM_182961.4:c.658−36G>A NM_174936.3:c.658−36G>A, and *APOB* NM_000384.3:c.12382G>A) between both cohorts. A novel variant of the *SYNE1* gene is associated with MI in young patients with FHpa.

## 1. Introduction

Cardiovascular diseases (CVD) are the leading cause of death globally. As the World Health Organization (WHO) reports, an estimated 17.9 million people died from CVD in 2019, representing 32% of all global deaths [1]. Ischemic heart disease (IHD) rose from the third to the first leading global cause of premature mortality, measured by years of life lost (YLL), between 1990 and 2019 [2]. In the EUROASPIRE V study, which involved people with confirmed coronary artery disease (CAD), up to 9% of the studied population were patients aged < 50 years [3]. According to the Framingham study, the MI incidence for a 10-year follow-up differs between the age groups—it was 12.9, 38.2, and 71.2 per 1000 in males and 2.2, 5.2, and 13.0 per 1000 in females in the age groups of 30–34, 35–44, and 45–54 years, respectively [4].

There are differences in risk factor profiles between individuals who experienced the first myocardial infarction at a very young (≤40 years) and at a young (age 41–50 years) age. When compared with young patients, very young individuals are more likely to use both marijuana and cocaine [5]. Conversely, patients aged 41 to 50 years are more likely to be diagnosed with hypertension or peripheral vascular disease, use alcohol, and have a higher 10-year atherosclerotic cardiovascular disease risk score. Nevertheless, the overall burden of traditional risk factors was similar between very young and young patients.

The INTERHEART study carried out in 52 countries around the world on more than 12 thousand patients showed smoking, lipid abnormalities, hypertension, and diabetes as more significant risk factors for MI in younger than in older patients [6]. Apart from those factors, a family history of premature IHD and genetic factors seem to be especially important in young people with MI.

According to the European Society of Cardiology guidelines for CVD prevention, the routine use of genetic risk scores is not yet recommended [7]. There is no consensus regarding which genes and corresponding single nucleotide polymorphisms should be included, and whether to use risk factor-specific or outcome-specific polygenic risk scores (PRS) [8]. The addition of a PRS for CAD to pooled cohort equations seems to be associated with a statistically significant, yet modest, improvement in the predictive accuracy for incidents of CAD and improved risk stratification for only a small proportion of individuals [9]. Among young adults from the CARDIA study (Coronary Artery Risk Development in Young Adults) and Framingham Offspring Study, PRS improved model discrimination for coronary atherosclerosis, but improvements were smaller than those associated with modifiable risk factors [10].

Nevertheless, there is no doubt that genetic information is a valuable complement to traditional factors in CAD risk stratification, particularly in young people. The wide use of genetic information to predict CAD development and incidence still requires investigation before clinical implementation. The group of special interest is a population of young people with CAD and a family history of premature IHD, defined as MI or ischemic stroke in first-degree relatives at age < 55 years in men and <65 years in women [11].

Thus, the aim of our study was to investigate the role of genetic variants in this very special population—young patients with MI and a family history of premature IHD. The involvement of genetic factors in the development of CAD in such patients is worthy of investigation due to at least two reasons. Firstly, a family history of premature CAD suggests the role of genetic background in the development of atherosclerosis in this group of patients. Secondly, the very young age of people suffering from MI forces us to look for genetic factors of the disease. Their recognition is crucial for the identification of young people with an early MI high risk and for the implementation of early cardiovascular prevention strategies.

## 2. Results

### 2.1. Anthropometric, Biochemical and Clinical Characteristics of Studied and Control Groups

The mean BMI in the studied group was 28.8 kg/m^2^ (SD ± 3.5), with a total cholesterol level 202.3 mg/dL (SD ± 3.5), HDL 40.6 mg/dL (SD ± 9.6), LDL 124.0 mg/dL (SD ± 28.9), triglycerides 164.4 mg/dL (SD ± 63.5), glucose 108.3 mg/dL (SD ± 19.9), and creatinine 0.9 mg/dL (SD ± 0.2); Table 1. Regarding other risk factors, 63 patients (90%) were smokers (mean 24.1 pack-years, SD ± 13.0), 44 (62.8%) had hypertension, 11 (15.7%) had diabetes mellitus, and 8 (11.4%) had depression.

The mean BMI in the control group was 24.5 kg/m^2^, (SD ± 4.8), with 176 smokers (29.5%). The mean BMI in the control group matched with a studied group by age and gender was 26.08 kg/m^2^ (SD ± 4.3), and this group included 62 smokers (29.5%).

The comparison of the studied group and control group is presented in Table 2.

### 2.2. Next-Generation Sequencing

The sequencing encompassed the coding sequences of 159 genes with 40 nucleotide intronic flanks based on the Genome Reference Consortium Human Build 37 (GRCh37) assembly. From the variants in the study group, low quality variants were removed. Subsequently, variants were narrowed down to those classified as variants of unknown significance, possibly pathogenic or pathogenic, according to the bioinformatic algorithm based on the American College of Medical Genetics and Genomics (ACMG) standards and guidelines [12]. This allowed for variants which were biologically insignificant to be filtered out. Furthermore, only variants of the allele count greater than or equal to 5 in the MI group were considered in further analysis, to minimize the risk of false positive findings. This resulted in the identification of four variants: *LDLR* (low-density lipoprotein receptor) gene NM_001195800.2:c.667G>A, *SYNE1* (spectrin-repeat-containing nuclear envelope protein 1) gene NM_182961.4:c.20396+22A>G, *PCSK9* (proprotein convertase subtilisin/kexin type 9) gene NM_174936.3:c.658–36G>A and *APOB* (apolipoprotein B) gene NM_000384.3:c.12382G>A. Allelic count of these SNPs (single nucleotide polymorphisms) was assessed in the control group. Furthermore, for each variant, the Hardy–Weinberg equilibrium was tested in the control population showing (*p* > 0.05). The allelic count of other SNPs with at least unknown significance identified in the study group were not greater than or equal to 5. Due to the low allelic count, Fisher’s exact test was chosen over a Chi-squared test to check for differences in allelic frequencies in both groups; see Table 3.

Sequencing and statistical analysis performed in the MI patients and healthy controls revealed the presence of a single variant in the *SYNE1* gene, with an enriched abundance within the patients. We did not observe any statistically significant differences between the frequency of other variants, including known pathogenic variants related to familial hypercholesterolemia (FH) as well as variants of unknown significance of FH-related genes: *LDLR* c.667G>A, *PCSK9* c.658–36G>A, and *APOB* c.12382G>A.

*SYNE1* gene variant rs36215567 (NM_182961.4: c.20396+22A>G) was statistically significantly more prevalent in the studied group in comparison to the control population with OR 4.80, 95%CI 1.43–14.45 (*p* = 0.005), as well as when compared to the control population matched by age and gender: OR 9.31, 95%CI 1.64–95.41(*p* = 0.004); see Figure 1.

Considering that the study group differed significantly in BMI and the number of smokers (*p* < 0.001) when compared with both an unmatched and matched control population, the multivariable logistic regression model was created. The model included smoking status, BMI and presence of *SYNE1* rs36215567. The model confirmed this SNP to be an independent risk factor for premature MI with OR 13.86 95%CI 1.64–117.45 (*p* = 0.016); see Table 4.

## 3. Discussion

The main aim of this study was to identify single variants predisposing to MI at a young age, as the genetic background seems to be particularly important in the pathogenesis of CAD in this group. In our previous study, we revealed that a younger age of patients with myocardial infarction is associated with a higher number of relatives with a history of premature MI/ischaemic stroke [13].

Numerous evidence indicates that CAD at a young age is associated with a genetic background. A significant enrichment of increased polygenic score has been noted among patients with early-onset (age ≤ 55 years) MI as compared with population-based controls, with median polygenic score among patients in the 72nd percentile [14]. Both familial hypercholesterolemia mutations and high polygenic score were associated with more than three-fold increased odds of early-onset MI. Monogenic familial hypercholesterolemia or a high genome-wide polygenic score confer to a 4- to 5-fold relative risk, which suggests that assessment for familial hypercholesterolemia mutations and genome-wide polygenic scores could be very useful for risk prediction, especially when combined with traditional risk factors for CAD [15].

Using data from the UK Biobank on 306,654 individuals without a history of CVD and not receiving lipid-lowering treatments, Sun L et al. calculated risk discrimination and reclassification upon addition of PRSs to risk factors in a conventional risk prediction model (i.e., age, sex, systolic blood pressure, smoking status, history of diabetes, and total and high-density lipoprotein cholesterol) [16]. The addition of information on PRSs increased the C-index by 0.012 (95% CI 0.009–0.015) and resulted in continuous net reclassification improvements—among people at intermediate (5% to <10%) 10-year CVD risk, it could help prevent one additional CVD event for approximately every 340 individuals screened. Such a targeted strategy could help prevent 7% more CVD events than conventional risk prediction alone. Moreover, lastly, it has been shown that adding PRS to clinical risk assessment has significantly improved the identification of people who experience a serious CVD event, especially at a young age—among people aged 40–54 years, clinical risk assessment alone identified 26.0% (95% CI: 16.5–37.6%) of those who developed a major CVD event, while the combination of clinical and genetic approaches identified 38.4% (95% CI: 27.2–50.5%) [17].

In this study, we used Next-Generation Sequencing to identify possible MI-related genetic variants in young patients. Since the familial aggregation of MI is an independent risk factor for the disease, we decided to perform an analysis using a panel of 159 genes related to various cardiovascular disorders, including 26 genes that are known to be involved in myopathies and dyslipidemias. Interestingly, our study did not show any statistically important difference in frequencies of variants commonly associated with dyslipidemias between the two studied groups [18]. We identified a single genetic variant in the *SYNE1* gene, whose prevalence was higher in MI patients. The rs36215567 (NM_182961.4: c.20396+22A>G) variant is located in an intron of a gene, possibly altering splicing of the *SYNE1* gene and resulting in altered activity of the translated protein, though the effect of this variant on the production of different *SYNE1* transcripts has not been described. It is a rare variant occurring at the frequency of 0.89% in the European population (1000 genomes project).

*SYNE1* (spectrin-repeat-containing nuclear envelope protein 1; Nesprin1) gene is a protein-coding gene located in a long arm of chromosome 6, position 6q25.2. Its protein is a member of the spectrin family of structural proteins that link the nuclear plasma membrane to the actin cytoskeleton. It is expressed in numerous tissues, but it has mainly been mapped to the aortic vascular smooth muscle cells and heart [19]. *SYNE1* aberrances were mainly associated with arthrogryposis and muscular dystrophies (Online Mendelian Inheritance in Men), but more recent papers have identified its involvement in dilated cardiomyopathy [20,21]. Interestingly, this association results mainly from the presence of alternatively spliced gene isoforms. *SYNE1* has been shown also to be involved in angiotensin-II-induced cardiac hypertrophy by the miR-525-5p/specific protein transcription factor SP1 axis [22].

Moreover, a recent study revealed an important role of *SYNE1* in cell regeneration and showed that its levels decrease with age, possibly leading to a decrease in cardiac function [23]. Thus, even though the exact role of the SYNE1 protein in cardiovascular homeostasis is still largely unknown, a growing amount of data suggest its role in the maintenance of the proper function of the cardiac and arterial muscles. Vascular smooth muscles as well as endothelial cells and macrophages produce interleukin 1 (Il-1). The inflammation processes, including the action of Il-1, are crucial but still among the multiple pathways of the heart microenvironment modifications involved in mechanisms of atherosclerosis [24]. A potential crosstalk between *SYNE1* and IL-1, IL-1β main receptor NOD-, LRR- and pyrin domain-containing protein 3 (NLRP3) or nuclear factor kappa-light-chain-enhancer of activated B cells (NF-kB) could be one of the possible explanations of the association of *SYNE1* and CAD shown in our study. Moreover, it opens new potential therapeutic opportunities for early prevention. It has been shown that some nutraceuticals are endowed with anti-inflammatory properties, exert cardioprotective activity, and protect against oxidative stress by their influence against Il-1 [25]. Apart from traditional treatment with statins or new drugs like PCSK-9 inhibitor, they may be a valuable option for early cardiovascular prevention in people with increased CAD risk based on the presence of the *SYNE1* variant.

There are also available data indicating a possible role of *SYNE1* variants (although others than those shown in our study) in the development of CAD by their involvement in the morbidity of hypercholesterolemia and hypertriglyceridemia through gene–gene and SNP-SNP interactions [26]. This suggests, considering the results of our study, other potential gene–gene interactions including *SYNE1* gene variants in the pathogenesis of CAD.

The main limitation of this study is the relatively small number of participants. The assumptions of the study, including young patients with MI and a family history of premature atherosclerosis, make this population very unique, but at the same time difficult to enlarge. On the other hand, the high homogeneity of the groups, limited to the Polish Caucasian population, may be important in the context of potential population and racial differences in the pathogenesis of CAD, especially when analyzing genetic risk factors. Moreover, this the first report regarding a potential role of *SYNE1* in premature CAD, which means that there are no similar data in wide databases yet, or functional studies carried out to elucidate potential mechanisms. However, this strengthens the value of our findings.

## 4. Material and Methods

### 4.1. Patients

The studied group consisted of 70 patients with MI aged 26–49 (mean 43.1, SD ± 4.3; 17 women and 53 men), admitted to the Department of Cardiology, Centre of Postgraduate Medical Education, Grochowski Hospital. All patients had been diagnosed with MI based on clinical presentation, ecg and biochemical criteria, including MI with ST elevation (STEMI) in 18 cases (25.7%) and MI without ST elevation (NSTEMI) in 52 cases (74.3%), and had a family history of premature atherosclerosis, defined as MI or ischemic stroke in first-degree relatives at age < 65 years in women or <55 years in men. All patients, except 2 (no consent), had undergone coronary angiography, revealing 1-vessel coronary artery disease (VCAD) in 33 patients (47.1%), 2-VCAD in 17 patients (24.3%), and 3-VCAD in 20 patients (28.6%); see Table 5. The mean left ventricular ejection fraction (LVEF) was 51.8% (SD ± 8.0).

There were 15 (21.4%) participants of the study with a university degree, 46 (65.7%) who had completed high school education and 7 (10%) who had completed elementary school education (2 patients did not provide information). Regarding type of job, there were 37 (52.8%) blue collar workers, 23 (32.9%) white collar workers, 3 (4.3%) who were unemployed and 2 (2.9%) people on pensions (5 patients did not provide information). Regarding marriage status, there were 40 (57.1%) married people in the study, 18 (25.7%) single, 6 (8.6%) divorced, 2 widows and 1 widower (4.3%); 3 patients did not provide information.

All participants of the study provided written informed consent. The Ethical Committee of the Centre of Postgraduate Medical Education approved the study protocol. The investigation conforms to the principles outlined in the Declaration of Helsinki.

The results of sequencing were compared to data from the reference Warsaw Genomics control population, consisting of 597 apparently healthy people without CVD, including a history of MI, (418 women, 179 men) aged 18–83 (mean 40.5, SD ± 12.4), as a whole and after matching using Propensity Score Matching with a studied group by age and gender in a proportion of 1:3 (210 people, 51 women and 159 men, aged 18–77, mean 42.1, SD ± 10.6).

### 4.2. Anthropometric, Clinical and Biochemical Analysis

BMI was calculated as weight (kg)/height (m^2^). Depression and smoking status, including duration and intensity (cigarette number per day) of smoking, were assessed based on the patient’s history. Hypertension was assessed based on the medical history and treatment or on a mean value of two measurements of systolic (SBP) and diastolic (DBP) blood pressure performed after at least 5 min sitting, in 5 min intervals. Hypertension was defined as values ≥ 140 mmHg SBP and/or ≥90 mmHg DBP according to ESH/ESC (European Society of Hypertension, European Society of Cardiology) guidelines. Diabetes mellitus was assessed based on the medical history and treatment or based on fasting plasma glucose ≥ 126 mg/dL or ≥200 mg/dL in an oral glucose tolerance test according to WHO and ADA (American Diabetes Association) guidelines.

Blood samples were collected on admission and the next morning. Biochemical analyses, including glucose, total HDL and LDL cholesterol and triglycerides (TG) plasma concentrations were performed in fasting blood samples through standard enzymatic methods using COBAS INTEGRA 800 regents and equipment (Roche Diagnostics Gmbh, Mannheim, Germany).

### 4.3. Genotyping

Total DNA was extracted from whole peripheral blood samples. The extraction was performed on a Maxwell^®^ RSC 48 Instrument (Promega, Madison, WI, USA) using the Maxwell^®^ RSC Blood DNA Kit, according to the protocol. The targeted enrichment library was prepared and analyzed using the Next-Generation Sequencing (NGS) method. The total DNA input for NGS libraries was 35 ng. The sample libraries were prepared on Biomed i7 (Beckam and Coulter) with the use of the Kapa HyperPlus protocol (Roche Diagnostics Gmbh, Mannheim, Germany). The concentration and quality of the libraries were subsequently verified on SpectraMax (Molecular Devices, San Jose, CA, USA) and TapeStation 4200 and subjected to the hybridization capture (NimbleGen EZ SeqCap, Choice kit, custom panel designed, Roche Diagnostics). The hybridization was conducted according to the Roche protocol. The size, concentration, and quality of the final NGS library obtained were confirmed by TapeStation 4200 and Quantus (Promega, Mascot, Australia). The samples were sequenced on the Hiseq4000 (Illumina), 2 × 150 cycles.

### 4.4. Statistical Analysis

Statistical analyses were performed using the R software package version 4.2.3 (http://www.r-project.org/, with usage of Hardy–Weinberg, MatchIt and stats packages). Risks associated with detected variants were evaluated using Fisher’s exact test based on the allelic frequencies of variants in both groups. Moreover, to account for the difference in the number of smokers and BMI between the study and control groups, a logistic regression model was created with BMI and smoking status as variables.

## 5. Conclusions

In conclusion, a novel variant of the *SYNE1* gene is associated with myocardial infarction in patients of a young age with a family history of premature atherosclerosis. Nevertheless, the wide use of such genetic information to predict CAD development and incidences still requires further investigation, including wider genetic and functional studies, before clinical implementation.

## Figures and Tables

**Figure 1 ijms-26-02244-f001:**
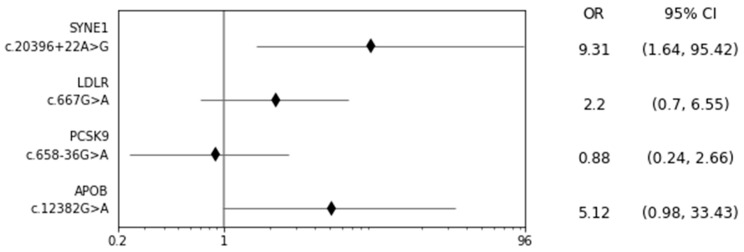
Incidence of *SYNE1* gene variants rs36215567 (NM_182961.4: c.20396+22A>G), *LDLR* c.667G>A, *PCSK9* c.658–36G>A, and *APOB* c.12382G>A in the studied group (70 patients) compared to the control population matched by age and gender (210 people); Fisher’s exact test. *APOB*—apolipoprotein B gene, CI—confidence intervals, G—guanine, *LDLR*—low-density lipoprotein receptor gene, OR—odds ratio, *PCSK9*—proprotein convertase subtilisin/kexin type 9 gene, SNP—single nucleotide polymorphism, *SYNE1*—spectrin-repeat-containing nuclear envelope protein 1 gene.

**Table 1 ijms-26-02244-t001:** Biochemical characteristics of studied group (70 patients). HDL—high density lipoprotein, LDL—low density lipoprotein, SD—standard deviation.

Studied Group Characteristics	Mean (±SD)
Total cholesterol mg/dL	202.3 (±3.5)
HDL mg/dL	40.6 (±9.6)
LDL mg/dL	124.0 (±28.9)
Triglycerides mg/dL	164.4 (±63.5)
Glucose mg/dL	108.3 (±19.9)
Creatinine mg/dL	0.9 (±0.2)

**Table 2 ijms-26-02244-t002:** CAD risk factors in studied group (70 patients) and control group (597 people). BMI—body mass index, CAD—coronary artery disease.

Risk Factors	Studied Group	Control Group
BMI kg/m^2^ mean (±SD)	28.8 (±3.5)	24.5 kg/m^2^
Number of affected patients—n (%)		
Smoking	63 (90%)	176 (29.5%)
Hypertension	44 (62.8%)	0
Diabetes mellitus	11 (15.7%)	0
Depression	8 (11.4%)	0

**Table 3 ijms-26-02244-t003:** Analyzed SNPs characteristics—number of genotypes (wild type/alternative heterozygote/alternative homozygote) in the studied group (70 patients) and in the control group unmatched (597 people) and matched by age and sex (210 people).

Gene	SNP (Genotypes)	MI Group	Control Group (Unmatched)	Control Group (Matched)
*SYNE1*	rs36215567 (AA/AG/GG)	64/6/0	586/11/0	208/2/0
*LDLR*	rs11669576 (GG/GA/AA)	61/7/0	541/48/1	198/10/0
*PCSK9*	rs11800265 (GG/GA/AA)	61/5/0	414/43/1	150/14/0
*APOB*	rs1801703 (GG/GA/AA)	65/5/0	587/9/0	207/3/0

A—adenine, *APOB*—apolipoprotein B gene, G—guanine, *LDLR*—low-density lipoprotein receptor gene, MI—myocardial infarction, *PCSK9*—proprotein convertase subtilisin/kexin type 9 gene, SNP—single nucleotide polymorphism, *SYNE1*—spectrin-repeat-containing nuclear envelope protein 1 gene.

**Table 4 ijms-26-02244-t004:** Multivariable regression model including smoking status, BMI and presence of *SYNE1* rs36215567 variant confirmed this SNP as an independent risk factor for premature MI; studied group—70 patients, control group (matched)—210 people, * statistically significant. BMI—body mass index, CI—confidence intervals, MI—myocardial infarction, OR—odds ratio, SNP—single nucleotide polymorphism, *SYNE1*—spectrin-repeat-containing nuclear envelope protein 1 gene.

Variable	OR	OR (95% CI)	*p* Value
BMI	1.13	1.05–1.23	0.001 *
Smoking status	19.40	8.19–45.94	<0.001 *
*SYNE1* rs36215567 variant	13.86	1.64–117.45	0.016 *

**Table 5 ijms-26-02244-t005:** Clinical and haemodynamic characteristics of the studied group (70 patients). CAD—coronary artery disease, LVEF—left ventricle ejection fraction, NSTEMI—non-ST elevation myocardial infarction, SD—standard deviation, STEMI—ST elevation myocardial infarction.

Clinical and Haemodynamic Characteristics	Number of Affected Patients n (%) Mean (±SD)
STEMI	18 (25.7%)
nSTEMI	52 (74.3%)
1-vessel CAD	33 (47.1%)
2-vessel CAD	17 (24.3%)
3-vessel CAD	20 (28.6%)
LVEF mean	51.8% (SD ± 8.0)

## Data Availability

The research data are available as a Appendix A to the manuscript as well as on line: https://figshare.com/articles/dataset/ijms-26-02244-supplementary/28512458?file=52687280, accessed on 20 February 2025.

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
