# Peer review of "Intronic SYNE1 Gene Novel Variant Associated with Myocardial Infarction in Young People with a Family History of Premature Atherosclerosis: A Case–Control Study in the Polish Population†"

_ijms, 2025, doi:10.3390/ijms26052244_

Round 1

Reviewer 1 Report

Comments and Suggestions for Authors

Title

Suggestion 1: Please consider including the type of study and the control population in the title and specify that the identified variant is intronic. Suggested title: 'Intronic SYNE1 Gene Novel Variant Associated with Myocardial Infarction in Young People with a Family History of Premature Atherosclerosis: A Case-Control Study Using the Warsaw Genomics Control Population.'

Abstract

Recommendation 1 - Please use the HGVS nomenclature to describe genetic variants throughout the abstract and the article. Before using a variant name, verify its correctness using tools like Mutalyzer - https://mutalyzer.nl/normalizer

Recommendation 2 - The headings 'Background,' 'Methods,' 'Results,' and 'Conclusion' are not required for publication in IJMS. Please remove them and adjust the abstract accordingly.

Introduction

Recommendation 3 - Please elaborate on the aim of the study at the end of the introduction, making it as specific and complete as possible.

Results

Recommendation 4 - Please include as many demographic, clinical, anthropometric, and biochemical characteristics as possible for the control group (reference Warsaw Genomics control population) and compare them with the study group. Present this comparison in a table, providing evidence that the control group was appropriate for the comparison. At least age is available, correct?

Recommendation 5 - Please clarify the criteria for selecting the SYNE1 rs36215567, LDLR rs11669576, PCSK9 rs11800265, and APOB rs1801703 polymorphisms. Were any rare or family-specific variants identified in the analyzed panel?

Discussion

No comments

Methods

Recommendation 6 - (Related to Recommendation 4) Please provide more details on the control population used (reference Warsaw Genomics control population) and explain how it relates to the study patients, demonstrating its appropriateness for comparison. It would be valuable to include a table comparing cases and controls in terms of common clinical and demographic variables to clearly show that the control population was suitable for comparison.

Recommendation 7 - Please provide the list of the 159 genes tested and specify which bioinformatics algorithms were used in the analysis. It is not possible to determine which genes were tested based on the current description. Additionally, include the full name of the NimbleGen EZ SeqCap panel (Roche Diagnostics) and clarify whether it was a custom or off-the-shelf panel. The bioinformatics analyses used were also not described, nor was the genomic reference version specified. This section is insufficient; please improve it by providing these details.

Recommendation 8 - Please provide a more detailed description of the statistical analyses, including the specific R packages used. Ensure that the description is sufficiently detailed to allow full reproducibility of the analysis. Additionally, if possible, include the final de-identified raw data as supplementary material for transparency and reproducibility. This should include genotype data and the variables used in the multivariable regression analysis to allow verification of the findings.

Conclusion

No comments

Reviewer 2 Report

Comments and Suggestions for Authors

Manuscript titled " SYNE1 gene novel variant is associated with myocardial infarction in young people with a family history of premature atherosclerosis." is a very interesting paper in the field of preventive cardiology. The overall structure is of good quality; methods and figures are clear and well performed. However, i can suggest to the authors to improve in two parts the manuscript:

1. First,  in introduction, authors should explain if SYNE1 interact with NLRP3 and IL1 and its involvement in atherosclerosis ( cite 10.26355/eurrev_202111_27124 )

2. Second, SYNE1 expression should be modulated with nutraceuticals and some drugs like PCSK9i. Please explain this point ( cite 10.1016/j.ijpharm.2015.08.039 )

The manuscrit should be accepted after minor revision. 

Round 2

Reviewer 1 Report

Comments and Suggestions for Authors

I have no additional comments. I congratulate the authors on their work and believe it is ready for publication.